# DDSP:
# DIFFERENTIABLE DIGITAL SIGNAL PROCESSING

**Jesse Engel, Lamtharn Hantrakul, Chenjie Gu, & Adam Roberts**
Google Research, Brain Team
Mountain View, CA 94043, USA
{jesseengel,hanoih,gcj,adarob}@google.com

## ABSTRACT

Most generative models of audio directly generate samples in one of two domains: time or frequency. While sufficient to express any signal, these representations are inefficient, as they do not utilize existing knowledge of how sound is generated and perceived. A third approach (vocoders/synthesizers) successfully incorporates strong domain knowledge of signal processing and perception, but has been less actively researched due to limited expressivity and difficulty integrating with modern auto-differentiation-based machine learning methods. In this paper, we introduce the Differentiable Digital Signal Processing (DDSP) library, which enables direct integration of classic signal processing elements with deep learning methods. Focusing on audio synthesis, we achieve high-fidelity generation without the need for large autoregressive models or adversarial losses, demonstrating that DDSP enables utilizing strong inductive biases without losing the expressive power of neural networks. Further, we show that combining interpretable modules permits manipulation of each separate model component, with applications such as independent control of pitch and loudness, realistic extrapolation to pitches not seen during training, blind dereverberation of room acoustics, transfer of extracted room acoustics to new environments, and transformation of timbre between disparate sources. In short, DDSP enables an interpretable and modular approach to generative modeling, without sacrificing the benefits of deep learning. The library is publicly available[1] and we welcome further contributions from the community and domain experts.

## 1 INTRODUCTION

Neural networks are universal function approximators in the asymptotic limit (Hornik et al., 1989), but their practical success is largely due to the use of strong structural priors such as convolution (Le-Cun et al., 1989), recurrence (Sutskever et al., 2014; Williams & Zipser, 1990; Werbos, 1990), and self-attention (Vaswani et al., 2017). These architectural constraints promote generalization and data efficiency to the extent that they align with the data domain. From this perspective, end-to-end learning relies on structural priors to scale, but the practitioner's toolbox is limited to functions that can be expressed differentiably. Here, we increase the size of that toolbox by introducing the Differentiable Digital Signal Processing (DDSP) library, which integrates interpretable signal processing elements into modern automatic differentiation software (TensorFlow). While this approach has broad applicability, we highlight its potential in this paper through exploring the example of audio synthesis.

Objects have a natural tendency to periodically vibrate. Small shape displacements are usually restored with elastic forces that conserve energy (similar to a canonical mass on a spring), leading to harmonic oscillation between kinetic and potential energy (Smith, 2010). Accordingly, human hearing has evolved to be highly sensitive to phase-coherent oscillation, decomposing audio into spectrotemporal responses through the resonant properties of the basilar membrane and tonotopic

---

[1]Online Resources:
Code: https://github.com/magenta/ddsp
Audio Examples: https://goo.gl/magenta/ddsp-examples
Colab Demo: https://goo.gl/magenta/ddsp-demo

mappings into the auditory cortex (Moerel et al., 2012; Chi et al., 2005; Theunissen & Elie, 2014). However, neural synthesis models often do not exploit this periodic structure for generation and perception.

## 1.1 Challenges of Neural Audio Synthesis

As shown in Figure 1, most neural synthesis models generate waveforms directly in the time domain, or from their corresponding Fourier coefficients in the frequency domain. While these representations are general and can represent any waveform, they are not free from bias. This is because they often apply a prior over generating audio with aligned wave packets rather than oscillations. For example, strided convolution models–such as SING (Defossez et al., 2018), MCNN (Arik et al., 2019), and WaveGAN (Donahue et al., 2019)–generate waveforms directly with overlapping frames. Since audio oscillates at many frequencies, all with different periods from the fixed frame hop size, the model must precisely align waveforms between different frames and learn filters to cover all possible phase variations. This challenge is visualized on the left of Figure 1.

Fourier-based models–such as Tacotron (Wang et al., 2017) and GANSynth (Engel et al., 2019)–also suffer from the phase-alignment problem, as the Short-time Fourier Transform (STFT) is a representation over windowed wave packets. Additionally, they must contend with spectral leakage, where sinusoids at multiple neighboring frequencies and phases must be combined to represent a single sinusoid when Fourier basis frequencies do not perfectly match the audio. This effect can be seen in the middle diagram of Figure 1.

Autoregressive waveform models–such as WaveNet (Oord et al., 2016), SampleRNN (Mehri et al., 2016), and WaveRNN (Kalchbrenner et al., 2018)–avoid these issues by generating the waveform a single sample at a time. They are not constrained by the bias over generating wave packets and can express arbitrary waveforms. However, they require larger and more data-hungry networks, as they do not take advantage of a bias over oscillation (size comparisons can be found in Table B.6). Furthermore, the use of teacher-forcing during training leads to exposure bias during generation, where errors with feedback can compound. It also makes them incompatible with perceptual losses such as spectral features (Defossez et al., 2018), pretrained models (Dosovitskiy & Brox, 2016), and discriminators (Engel et al., 2019). This adds further inefficiency to these models, as a waveform's shape does not perfectly correspond to perception. For example, the three waveforms on the right of Figure 1 sound identical (a relative phase offset of the harmonics) but would present different losses to an autoregressive model.

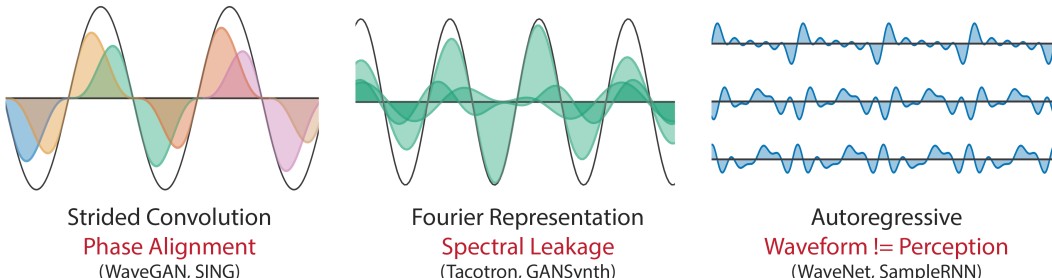

Figure 1: Challenges of neural audio synthesis. Full description provided in Section 1.1.

## 1.2 Oscillator Models

Rather than predicting waveforms or Fourier coefficients, a third model class directly generates audio with oscillators. Known as vocoders or synthesizers, these models are physically and perceptually motivated and have a long history of research and applications (Beauchamp, 2007; Morise et al., 2016). These "analysis/synthesis" models use expert knowledge and hand-tuned heuristics to extract synthesis parameters (analysis) that are interpretable (loudness and frequencies) and can be used by the generative algorithm (synthesis).

Neural networks have been used previously to some success in modeling pre-extracted synthesis parameters (Blaauw & Bonada, 2017; Chandna et al., 2019), but these models fall short of end-to-end learning. The analysis parameters must still be tuned by hand and gradients cannot flow through the synthesis procedure. As a result, small errors in parameters can lead to large errors in

the audio that cannot propagate back to the network. Crucially, the realism of vocoders is limited by the expressivity of a given analysis/synthesis pair.

## 1.3 CONTRIBUTIONS

In this paper, we overcome the limitations outlined above by using the DDSP library to implement fully differentiable synthesizers and audio effects. DDSP models combine the strengths of the above approaches, benefiting from the inductive bias of using oscillators, while retaining the expressive power of neural networks and end-to-end training.

We demonstrate that models employing DDSP components are capable of generating high-fidelity audio without autoregressive or adversarial losses. Further, we show the interpretability and modularity of these models enable:

- Independent control over pitch and loudness during synthesis.
- Realistic extrapolation to pitches not seen during training.
- Blind dereverberation of audio through seperate modelling of room acoustics.
- Transfer of extracted room acoustics to new environments.
- Timbre transfer between disparate sources, converting a singing voice into a violin.
- Smaller network sizes than comparable neural synthesizers.

Audio samples for all examples and figures are provided in the online supplement[2]. We *highly* encourage readers to listen to the samples as part of reading the paper.

## 2 RELATED WORK

**Vocoders.** Vocoders come in several varieties. Source-filter/subtractive models are inspired by the human vocal tract and dynamically filter a harmonically rich source signal (Flanagan, 2013), while sinusoidal/additive models generate sound as the combination of a set of time-varying sine waves (McAulay & Quatieri, 1986; Serra & Smith, 1990). Additive models are strictly more expressive than subtractive models but have more parameters as each sinusoid has its own time-varying loudness and frequency. This work builds a differentiable synthesizer off the Harmonic plus Noise model (Serra & Smith, 1990; Beauchamp, 2007): an additive synthesizer combines sinusoids in harmonic (integer) ratios of a fundamental frequency alongside a time-varying filtered noise signal.

**Synthesizers.** A separate thread of research has tried to estimate parameters for commercial synthesizers using gradient-free methods (Huang et al., 2014; Hoffman & Cook, 2006). Synthesizer outputs modeled with a variational autoencoder were recently used as a "world model" (Ha & Schmidhuber, 2018) to pass approximate gradients to a controller during learning (Esling et al., 2019). DDSP differs from black-box approaches to modeling existing synthesizers; it is a toolkit of differentiable DSP components for end-to-end learning.

**Neural Source Filter (NSF).** Perhaps closest to this work, promising speech synthesis results were recently achieved using a differentiable waveshaping synthesizer (Wang et al., 2019). The NSF can be seen as a specific DDSP model, that uses convolutional waveshaping of a sinusoidal oscillator to create harmonic content, rather than additive synthesis explored in this work. Both works also generate audio in the time domain and impose multi-scale spectrograms losses in the frequency domain. A key contribution of this work is to highlight how these models are part of a common family of techniques and to release a modular library that makes them accessible by leveraging automatic differentiation to easily mix and match components at a high level.

## 3 DDSP COMPONENTS

Many DSP operations can be expressed as functions in modern automatic differentiation software. We express core components as feedforward functions, allowing efficient implementation on parallel

---

[2]`https://goo.gl/magenta/ddsp`

hardware such as GPUs and TPUs, and generation of samples during training. These components include *oscillators*, *envelopes*, and *filters* (linear-time-varying finite-impulse-response, LTV-FIR). [3]

## 3.1 SPECTRAL MODELING SYNTHESIS

Here, as an example DDSP model, we implement a differentiable version of Spectral Modeling Synthesis (SMS) Serra & Smith (1990). This model generates sound by combining an additive synthesizer (adding together many sinusoids) with a subtractive synthesizer (filtering white noise). We choose SMS because, despite being parametric, it is a highly expressive model of sound, and has found widespread adoption in tasks as diverse as spectral morphing, time stretching, pitch shifting, source separation, transcription, and even as a general purpose audio codec in MPEG-4 (Tellman et al., 1995; Klapuri et al., 2000; Purnhagen & Meine, 2000).

As we only consider monophonic sources in these experiments, we use the Harmonic plus Noise model, that further constrains sinusoids to be integer multiples of a fundamental frequency (Beauchamp, 2007). One of the reasons that SMS is more expressive than many other parametric models because it has so many more parameters. For example, in the 4 seconds of 16kHz audio in the datasets considered here, the synthesizer coefficients actually have $\sim$2.5 times more dimensions than the audio waveform itself ((1 amplitude + 100 harmonics + 65 noise band magnitudes) * 1000 timesteps = 165,000 dimensions, vs. 64,000 audio samples). This makes them amenable to control by a neural network, as it would be difficult to realistically specify all these parameters by hand.

## 3.2 HARMONIC OSCILLATOR / ADDITIVE SYNTHESIZER

At the heart of the synthesis techniques explored in this paper is the sinusoidal oscillator. A bank of oscillators that outputs a signal $x(n)$ over discrete time steps, $n$, can be expressed as:

$$x(n) = \sum_{k=1}^{K} A_k(n) \sin(\phi_k(n)), \tag{1}$$

where $A_k(n)$ is the time-varying amplitude of the $k$-th sinusoidal component and $\phi_k(n)$ is its instantaneous phase. The phase $\phi_k(n)$ is obtained by integrating the instantaneous frequency $f_k(n)$:

$$\phi_k(n) = 2\pi \sum_{m=0}^{n} f_k(m) + \phi_{0,k}, \tag{2}$$

where $\phi_{0,k}$ is the initial phase that can be randomized, fixed, or learned.

For a harmonic oscillator, all the sinusoidal frequencies are harmonic (integer) multiples of a fundamental frequency, $f_0(n)$, i.e., $f_k(n) = kf_0(n)$, Thus the output of the harmonic oscillator is entirely parameterized by the time-varying fundamental frequency $f_0(n)$ and harmonic amplitudes $A_k(n)$. To aid interpretablity we further factorize the harmonic amplitudes:

$$A_k(n) = A(n)c_k(n). \tag{3}$$

into a global amplitude $A(n)$ that controls the loudness and a normalized distribution over harmonics $c(n)$ that determines spectral variations, where $\sum_{k=0}^{K} c_k(n) = 1$ and $c_k(n) \geq 0$. We also constrain both amplitudes and harmonic distribution components to be positive through the use of a modified sigmoid nonlinearity as described in the appendix. Figure 6 provides a graphical example of the additive synthesizer. Audio is provided in our online supplement[2].

## 3.3 ENVELOPES

The oscillator formulation above requires time-varying amplitudes and frequencies at the audio sample rate, but our neural networks operate at a slower frame rate. For instantaneous frequency upsampling, we found bilinear interpolation to be adequate. However, the amplitudes and harmonic distributions of the additive synthesizer required smoothing to prevent artifacts. We are able to achieve

---

[3]We have implemented further components such as wavetable synthesizers and non-sinusoidal oscillators, but focus here on components used in the experiments and leave the rest as future work.

this with a smoothed amplitude envelope by adding overlapping Hamming windows at the center of each frame and scaled by the amplitude. For these experiments we found a 4ms (64 timesteps) hop size and 8 ms frame size (50% overlap) to be responsive to changes while removing artifacts.

### 3.4 FILTER DESIGN: FREQUENCY SAMPLING METHOD

Linear filter design is a cornerstone of many DSP techniques. Standard convolutional layers are equivalent to linear time invariant finite impulse response (LTI-FIR) filters. However, to ensure interpretability and prevent phase distortion, we employ the frequency sampling method to convert network outputs into impulse responses of linear-phase filters.

Here, we design a neural network to predict the frequency-domain transfer functions of a FIR filter for every output frame. In particular, the neural network outputs a vector $H_l$ (and accordingly $h_l = \text{IDFT}(H_l)$) for the $l$-th frame of the output. We interpret $H_l$ as the frequency-domain transfer function of the corresponding FIR filter. We therefore implement a time-varying FIR filter.

To apply the time-varying FIR filter to the input, we divide the audio into non-overlapping frames $x_l$ to match the impulse responses $h_l$. We then perform frame-wise convolution via multiplication of frames in the Fourier domain: $Y_l = H_l X_l$ where $X_l = \text{DFT}(x_l)$ and $Y_l = \text{DFT}(y_l)$ is the output. We recover the frame-wise filtered audio, $y_l = \text{IDFT}(Y_l)$, and then overlap-add the resulting frames with the same hop size and rectangular window used to originally divide the input audio. The hop size is given by dividing the audio into equally spaced frames for each frame of conditioning. For 64000 samples and 250 frames, this corresponds to a hop size of 256.

In practice, we do not use the neural network output directly as $H_l$. Instead, we apply a window function $W$ on the network output to compute $H_l$. The shape and size of the window can be decided independently to control the time-frequency resolution trade-off of the filter. In our experiments, we default to a Hann window of size 257. Without a window, the resolution implicitly defaults to a rectangular window which is not ideal for many cases. We take care to shift the IR to zero-phase (symmetric) form before applying the window and revert to causal form before applying the filter.

### 3.5 FILTERED NOISE / SUBTRACTIVE SYNTHESIZER

Natural sounds contain both harmonic and stochastic components. The Harmonic plus Noise model captures this by combining the output of an additive synthesizer with a stream of filtered noise (Serra & Smith, 1990; Beauchamp, 2007). We are able to realize a differentiable filtered noise synthesizer by simply applying the LTV-FIR filter from above to a stream of uniform noise $Y_l = H_l N_l$ where $N_l$ is the DFT of uniform noise in domain [-1, 1].

### 3.6 REVERB: LONG IMPULSE RESPONSES

Room reverbation (reverb) is an essential characteristic of realistic audio, which is usually implicitly modeled by neural synthesis algorithms. In contrast, we gain interpretability by explicitly factorizing the room acoustics into a post-synthesis convolution step. A realistic room impulse response (IR) can be as long as several seconds, corresponding to extremely long convolutional kernel sizes (~10-100k timesteps). Convolution via matrix multiplication scales as $\mathcal{O}(n^3)$, which is intractable for such large kernel sizes. Instead, we implement reverb by explicitly performing convolution as multiplication in the frequency domain, which scales as $\mathcal{O}(n \log n)$ and does not bottleneck training.

## 4 EXPERIMENTS

For empirical verification of this approach, we test two DDSP autoencoder variants–supervised and unsupervised–on two different musical datasets: NSynth (Engel et al., 2017) and a collection of solo violin performances. The supervised DDSP autoencoder is conditioned on fundamental frequency (F0) and loudness features extracted from audio, while the unsupervised DDSP autoencoder learns F0 jointly with the rest of the network.

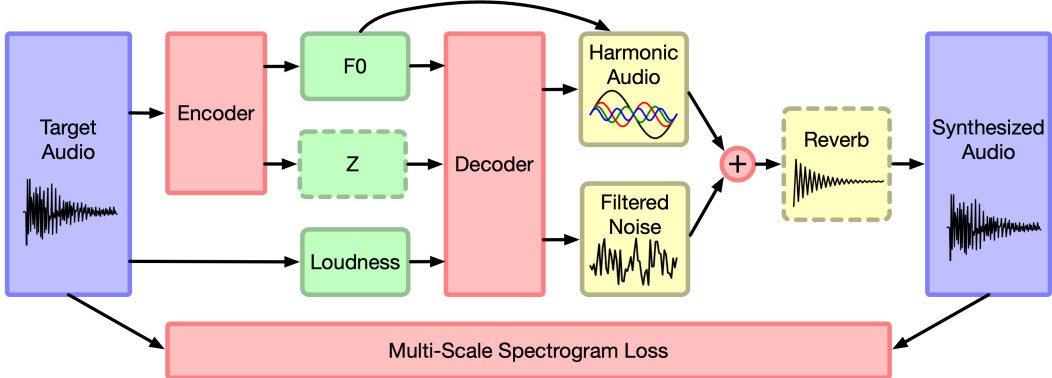

Figure 2: Autoencoder architecture. Red components are part of the neural network architecture, green components are the latent representation, and yellow components are deterministic synthesizers and effects. Components with dashed borders are not used in all of our experiments. Namely, $z$ is not used in the model trained on solo violin, and reverb is not used in the models trained on NSynth. See the appendix for more detailed diagrams of the neural network components.

## 4.1 DDSP AUTOENCODER

DDSP components do not put constraints on the choice of generative model (GAN, VAE, Flow, etc.), but we focus here on a deterministic autoencoder to investigate the strength of DDSP components independent of any particular approach to adversarial training, variational inference, or Jacobian design. Just as autoencoders utilizing convolutional layers outperform fully-connected autoencoders on images, we find DDSP components are able to dramatically improve autoencoder performance in the audio domain. Introducing stochastic latents (such as in GAN, VAE, and Flow models) will likely further improve performance, but we leave that to future work as it is orthogonal to the core question of DDSP component performance that we investigate in this paper.

In a standard autoencoder, an encoder network $f_{enc}(\cdot)$ maps the input $x$ to a latent representation $z = f_{enc}(x)$ and a decoder network $f_{dec}(\cdot)$ attempts to directly reconstruct the input $\hat{x} = f_{dec}(z)$. Our architecture (Figure 2) contrasts with this approach through the use of DDSP components and a decomposed latent representation.

**Encoders:** Detailed descriptions of the encoders are given in Section B.1. For the supervised autoencoder, the loudness $l(t)$ is extracted directly from the audio, a pretrained CREPE model with fixed weights (Kim et al., 2018) is used as an $f(t)$ encoder to extact the fundamental frequency, and optional encoder extracts a time-varying latent encoding $z(t)$ of the residual information. For the $z(t)$ encoder, MFCC coefficients (30 per a frame) are first extracted from the audio, which correspond to the smoothed spectral envelope of harmonics (Beauchamp, 2007), and transformed by a single GRU layer into 16 latent variables per a frame.

For the unsupervised autoencoder, the pretrained CREPE model is replaced with a Resnet architecture (He et al., 2016) that extracts $f(t)$ from a mel-scaled log spectrogram of the audio, and is jointly trained with the rest of the network.

**Decoder:** A detailed description of the decoder network is given in Section B.2. The decoder network maps the tuple $(f(t), l(t), z(t))$ to control parameters for the additive and filtered noise synthesizers described in Section 3. The synthesizers generate audio based on these parameters, and a reconstruction loss between the synthesized and original audio is minimized. The network architecture is chosen to be fairly generic (fully connected, with a single recurrent layer) to demonstrate that it is the DDSP components, and not other modeling decisions, that enables the quality of the work.

Also unique to our approach, the latent $f(t)$ is fed directly to the additive synthesizer as it has structural meaning for the synthesizer outside the context of any given dataset. As shown later in Section 5.2, this disentangled representation enables the model to both interpolate within and ex-

|  | Loudness ($L_1$) | F0 ($L_1$) | F0 Outliers |
|---|:---:|:---:|:---:|
| **Supervised** | | | |
| WaveRNN (Hantrakul et al., 2019) | 0.10 | 1.00 | 0.07 |
| DDSP Autoencoder | **0.07** | **0.02** | **0.003** |
| **Unsupervised** | | | |
| DDSP Autoencoder | 0.09 | 0.80 | 0.04 |

Table 1: Resynthesis accuracies. Comparison of DDSP models to SOTA WaveRNN model provided the same conditioning information. The supervised DDSP Autoencoder and WaveRNN models use the fundamental frequency from a pretrained CREPE model, while the unsupervised DDSP autoencoder learns to infer the frequency from the audio during training.

trapolate outside the data distribution. Indeed, recent work support incorporation of strong inductive biases as a prerequisite for learning disentangled representations (Locatello et al., 2018).

**Model Size:** Table B.6, compares parameter counts for the DDSP models and comparable models including GANSynth (Engel et al., 2019), WaveRNN (Hantrakul et al., 2019), and a WaveNet Autoencoder (Engel et al., 2017). The DDSP models have the fewest parameters (up to 10 times less), despite no effort to minimize the model size in these experiments. Initial experiments with very small models (240k parameters, 300x smaller than a WaveNet Autoencoder) have less realistic outputs than the full models, but still have fairly high quality and are promising for low-latency applications, even on CPU or embedded devices. Audio samples are available in the online supplement[2].

## 4.2 DATASETS

**NSynth:** We focus on a smaller subset of the NSynth dataset (Engel et al., 2017) consistent with other work (Engel et al., 2019; Hantrakul et al., 2019). It totals 70,379 examples comprised mostly of strings, brass, woodwinds and mallets with pitch labels within MIDI pitch range 24-84. We employ a 80/20 train/test split shuffling across instrument families. For the NSynth experiments, we use the autoencoder as described above (*with* the $z(t)$ encoder). We experiment with both the supervised and unsupervised variants.

**Solo Violin:** The NSynth dataset does not capture aspects of a real musical performance. Using the MusOpen royalty free music library, we collected 13 minutes of expressive, solo violin performances[4]. We purposefully selected pieces from a single performer (John Garner), that were monophonic and shared a consistent room environment to encourage the model to focus on performance. Like NSynth, audio is converted to mono 16kHz and divided into 4 second training examples (64000 samples total). Code to process the audio files into a dataset is available online.[5]

For the solo violin experiments, we use the supervised variant of the autoencoder (*without* the $z(t)$ encoder), and add a reverb module to the signal processor chain to account for room reverberation. While the room impulse response could be produced as an output of the decoder, given that the solo violin dataset has a single acoustic environment, we use a single fixed variable (4 second reverb corresponding to 64000 dimensions) for the impulse response.

### 4.2.1 MULTI-SCALE SPECTRAL LOSS

The primary objective of the autoencoder is to minimize reconstruction loss. However, for audio waveforms, point-wise loss on the raw waveform is not ideal, as two perceptually identical audio samples may have distinct waveforms, and point-wise similar waveforms may sound very different.

Instead, we use a multi-scale spectral loss–similar to the multi-resolution spectral amplitude distance in Wang et al. (2019)–defined as follows. Given the original and synthesized audio, we compute their (magnitude) spectrogram $S_i$ and $\hat{S}_i$, respectively, with a given FFT size $i$, and define the loss as the

---

[4]Five pieces by John Garner (II. Double, III. Corrente, IV. Double Presto, VI. Double, VIII. Double) from `https://musopen.org/music/13574-violin-partita-no-1-bwv-1002/`

[5]`https://github.com/magenta/ddsp`

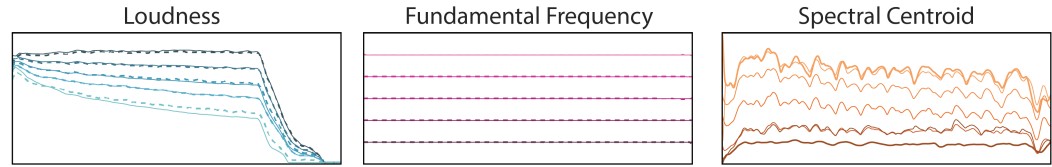

Figure 3: Separate interpolations over loudness, pitch, and timbre. The conditioning features (solid lines) are extracted from two notes and linearly mixed (dark to light coloring). The features of the resynthsized audio (dashed lines) closely follow the conditioning. On the right, the latent vectors, $z(t)$, are interpolated, and the spectral centroid of resulting audio (thin solid lines) smoothly varies between the original samples (dark solid lines).

sum of the L1 difference between $S_i$ and $\hat{S}_i$ as well as the L1 difference between $\log S_i$ and $\log \hat{S}_i$.

$$L_i = ||S_i - \hat{S}_i||_1 + \alpha || \log S_i - \log \hat{S}_i||_1. \tag{4}$$

where $\alpha$ is a weighting term set to 1.0 in our experiments. The total reconstruction loss is then the sum of all the spectral losses, $L_{\text{reconstruction}} = \sum_i L_i$. In our experiments, we used FFT sizes (2048, 1024, 512, 256, 128, 64), and the neighboring frames in the Short-Time Fourier Transform (STFT) overlap by 75%. Therefore, the $L_i$'s cover differences between the original and synthesized audios at different spatial-temporal resolutions.

## 5 RESULTS

### 5.1 HIGH-FIDELITY SYNTHESIS

As shown in Figure 5, the DDSP autoencoder learns to very accurately resynthesize the solo violin dataset. Again, we highly encourage readers to listen to the samples provided in the online supplement[2]. A full decomposition of the components is provided Figure 5. High-quality neural audio synthesis has previously required very large autoregressive models (Oord et al., 2016; Kalchbrenner et al., 2018) or adversarial loss functions (Engel et al., 2019). While amenable to an adversarial loss, the DDSP autoencoder achieves these results with a straightforward L1 spectrogram loss, a small amount of data, and a relatively simple model. This demonstrates that the model is able to efficiently exploit the bias of the DSP components, while not losing the expressive power of neural networks.

For the NSynth dataset, we quantitatively compare the quality of DDSP resynthesis with that of a state-of-the-art baseline using WaveRNN (Hantrakul et al., 2019). The models are comparable as they are trained on the same data, provided the same conditioning, and both targeted towards realtime synthesis applications. In Table 1, we compute loudness and fundamental frequency (F0) $L_1$ metrics described in Section C of the appendix. Despite the strong performance of the baseline, the supervised DDSP autoencoder still outperforms it, especially in F0 $L_1$. This is not unexpected, as the additive synthesizer directly uses the conditioning frequency to synthesize audio.

The unsupervised DDSP autoencoder must learn to infer its own F0 conditioning signal directly from the audio. As described in Section B.4, we improve optimization by also adding a perceptual loss in the form of a pretrained CREPE network (Kim et al., 2018). While not as accurate as the supervised DDSP version, the model does a fair job at learning to generate sounds with the correct frequencies without supervision, outperforming the supervised WaveRNN model.

### 5.2 INDEPENDENT CONTROL OF LOUDNESS AND PITCH

**Interpolation:** Interpretable structure allows for independent control over generative factors. Each component of the factorized latent variables $(f(t), l(t), z(t))$ independently alters samples along a matching perceptual axis. For example, Figure 3 shows an interpolation between two sound in the loudness conditioning $l(t)$. With other variables held constant, loudness of the synthesized audio closely matches the interpolated input. Similarly, the model reliably matches intermediate pitches between a high pitched $f(t)$ and low pitched $f(t)$. In Table C.2 of the appendix, we quantitatively demonstrate how across interpolations, conditioning independently controls the corresponding characteristics of the audio.

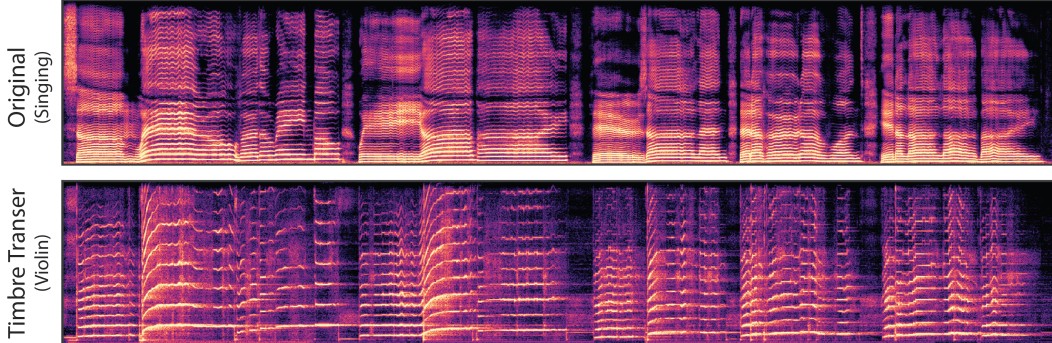

Figure 4: Timbre transfer from singing voice to violin. F0 and loudness features are extracted from the voice and resynthesized with a DDSP autoencoder trained on solo violin.

With loudness and pitch explicitly controlled by $(f(t), l(t))$, the model should use the residual $\boldsymbol{z}(t)$ to encode timbre. Although architecture and training do not strictly enforce this encoding, we qualitatively demonstrate how varying $\boldsymbol{z}$ leads to a smooth change in timbre. In Figure 3, we use the smooth shift in spectral centroid, or "center of mass" of a spectrum, to illustrate this behavior.

**Extrapolation:** As described in Section 4.1, $f(t)$ directly controls the additive synthesizer and has structural meaning outside the context of any given dataset. Beyond interpolating between datapoints, the model can extrapolate to new conditions not seen during training. The rightmost plot of Figure 7 demonstrates this by resynthesizing a clip of solo violin after shifting $f(t)$ down an octave and outside the range of the training data. The audio remains coherent and resembles a related instrument such as a cello. $f(t)$ is only modified for the synthesizer, as the decoder is still bounded by the nearby distribution of the training data and produces unrealistic harmonic content if conditioned far outside that distribution.

### 5.3    DEREVERBERATION AND ACOUSTIC TRANSFER

Removing reverb in a "blind" setting, where only reverberated audio is available, is a standing problem in acoustics (Naylor & Gaubitch, 2010). However, a benefit of our modular approach to generative modeling is that it becomes possible to completely separate the source audio from the effect of the room. For the solo violin dataset, the DDSP autoencoder is trained with an additional reverb module as shown in Figure 2 and described in Section 3.6. Figure 7 (left) demonstrates that bypassing the reverb module during resynthesis results in completely dereverberated audio, similar to recording in an anechoic chamber. The quality of the approach is limited by the underlying generative model, which is quite high for our autoencoder. Similarly, Figure 7 (center) demonstrates that we can also apply the learned reverb model to new audio, in this case singing, and effectively transfer the acoustic environment of the solo violin recordings.

### 5.4    TIMBRE TRANSFER

Figure 4 demonstrates timbre transfer, converting the singing voice of an author into a violin. F0 and loudness features are extracted from the singing voice and the DDSP autoencoder trained on solo violin used for resynthesis. To better match the conditioning features, we first shift the fundamental frequency of the singing up by two octaves to fit a violin's typical register. Next, we transfer the room acoustics of the violin recording (as described in Section 5.3) to the voice before extracting loudness, to better match the loudness contours of the violin recordings. The resulting audio captures many subtleties of the singing with the timbre and room acoustics of the violin dataset. Note the interesting "breathing" artifacts in the silence corresponding to unvoiced syllables from the singing.

## 6    CONCLUSION

The DDSP library fuses classical DSP with deep learning, providing the ability to take advantage of strong inductive biases without losing the expressive power of neural networks and end-to-end

learning. We encourage contributions from domain experts and look forward to expanding the scope of the DDSP library to a wide range of future applications.

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

# A  APPENDIX

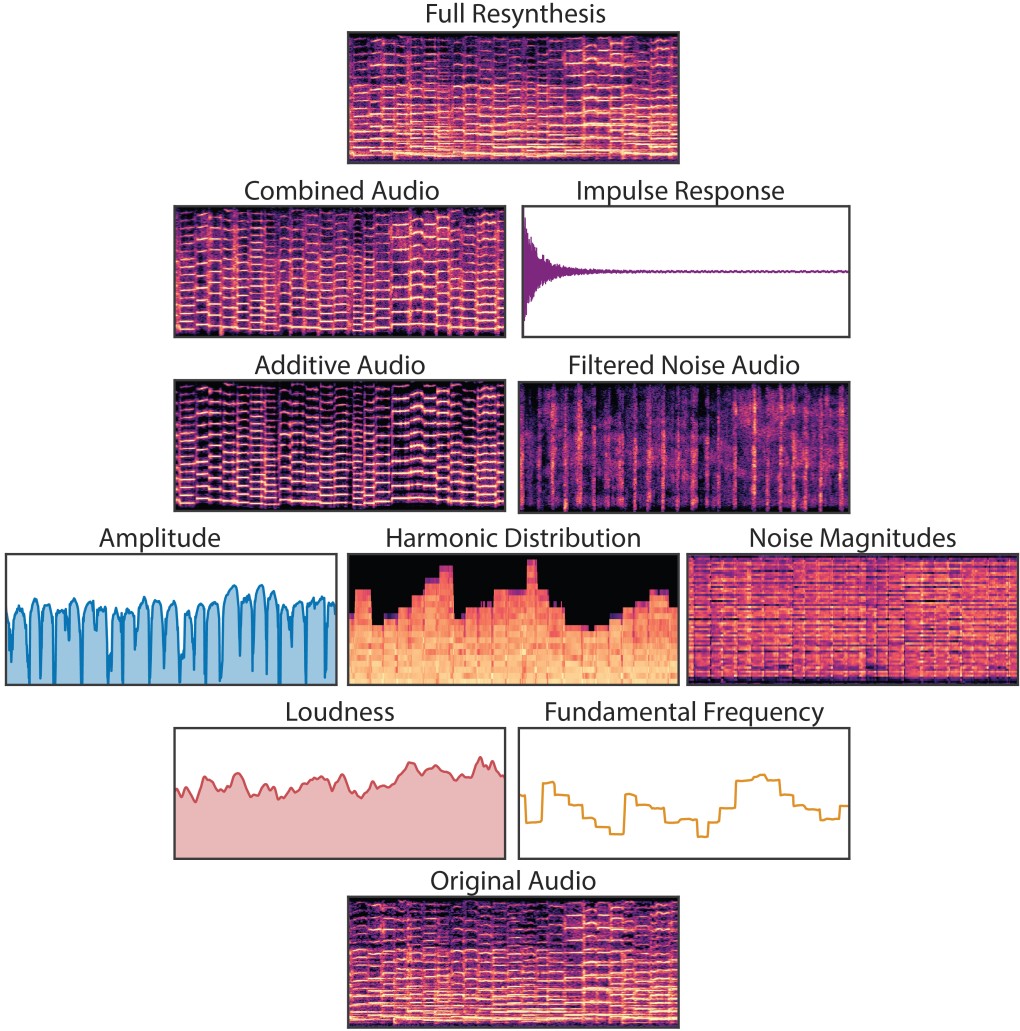

Figure 5: Decomposition of a clip of solo violin. Audio is visualized with log magnitude spectrograms. Loudness and fundamental frequency signals are extracted from the original audio. The loudness curve does not exhibit clear note segmentations because of the effects of the room acoustics. The DDSP autoencoder takes those conditioning signals and predicts amplitudes, harmonic distributions, and noise magnitudes. Note that the amplitudes are clearly segmented along note boundaries without supervision and that the harmonic and noise distributions are complex and dynamic despite the simple conditioning signals. Finally, the extracted impulse response is applied to the combined audio from the synthesizers to give the full resynthesis audio.

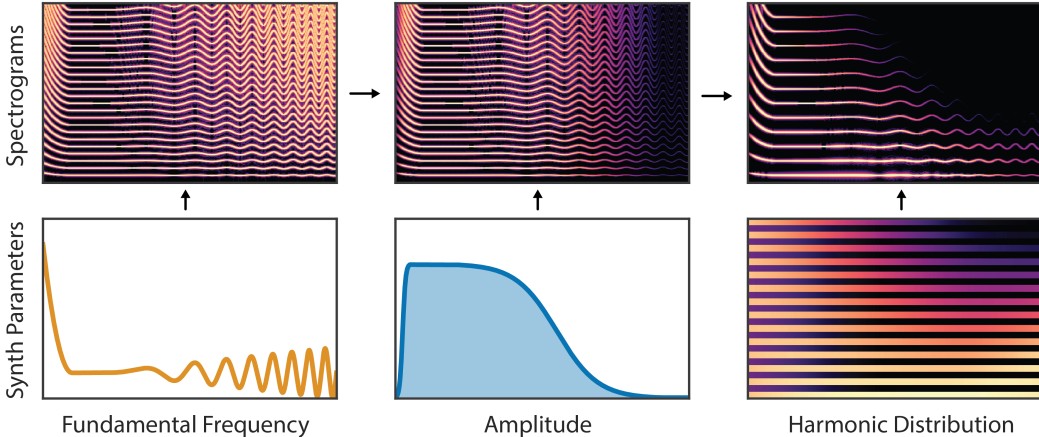

Figure 6: Diagram of the Additive Synthesizer component. The synthesizer generates audio as a sum of sinusoids at harmonic (integer) multiples of the fundamental frequency. The neural network is then tasked with emitting time-varying synthesizer parameters (fundamental frequency, amplitude, harmonic distribution). In this example linear-frequency log-magnitude spectrograms show how the harmonics initially follow the frequency contours of the fundamental. We then factorize the harmonic amplitudes into an overall amplitude envelope that controls the loudness, and a normalized distribution among the different harmonics that determines spectral variations.

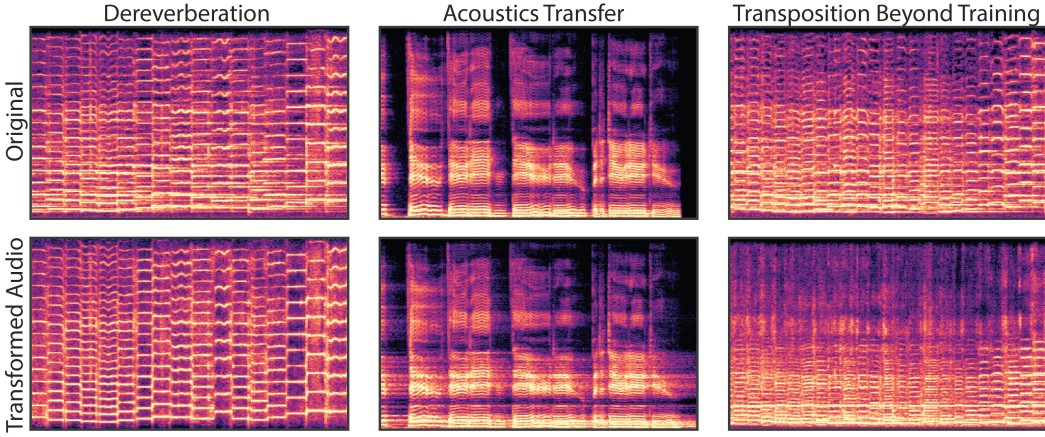

Figure 7: Interpretable generative models enables disentanglement and extrapolation. Spectrograms of audio examples include dereverberation of a clip of solo violin playing (left), transfer of the extracted room response to new audio (center), and transposition below the range of training data (right).

## B  MODEL DETAILS

### B.1  ENCODERS

The model has three encoders: $f$-encoder that outputs fundamental frequency $f(t)$, $l$-encoder that outputs loudness $l(t)$, and a $z$-encoder that outputs residual vector $z(t)$.

$f$**-encoder**: We use a pretrained CREPE pitch detector (Kim et al., 2018) as the $f$-encoder to extract ground truth fundamental frequencies (F0) from the audio. We used the "large" variant of CREPE, which has SOTA accuracy for monophonic audio samples of musical instruments. For our supervised autoencoder experiments, we fixed the weights of the $f$-encoder like (Hantrakul et al., 2019), and for our unsupervised autoencoder experiemnts we jointly learn the weights of a resnet model fed log mel spectrograms of the audio. Full details of the resnet architecture are show in Table 2.

$l$-**encoder**: We use identical computational steps to extract loudness as (Hantrakul et al., 2019). Namely, an A-weighting of the power spectrum, which puts greater emphasis on higher frequencies, followed by log scaling. The vector is then centered according to the mean and standard deviation of the dataset.

$z$-**encoder**: As shown in Figure 8, the encoder first calculates MFCC's (Mel Frequency Cepstrum Coefficients) from the audio. MFCC is computed from the log-mel-spectrogram of the audio with a FFT size of 1024, 128 bins of frequency range between 20Hz to 8000Hz, overlap of 75%. We use only the first 30 MFCCs that correspond to a smoothed spectral envelope. The MFCCs are then passed through a normalization layer (which has learnable shift and scale parameters) and a 512-unit GRU. The GRU outputs (over time) fed to a 512-unit linear layer to obtain $z(t)$. The $z$ embedding reported in this model has 16 dimensions across 250 time-steps.

| Residual Block | Output Size | $k_{Time}$ | $k_{Freq}$ | $s_{Freq}$ | $k_{Filters}$ |
|---|---|---|---|---|---|
| layer norm + relu | - | - | - | - | - |
| conv | - | 1 | 1 | 1 | $k_{Filters}/4$ |
| layer norm + relu | - | - | - | - | - |
| conv | - | 3 | 3 | $s_{Freq}$ | $k_{Filters}/4$ |
| layer norm + relu | - | - | - | - | - |
| conv | - | 1 | 1 | 1 | $k_{Filters}$ |
| add residual | - | - | - | - | - |
| Resnet | Output Size | $k_{Time}$ | $k_{Freq}$ | $s_{Freq}$ | $k_{Filters}$ |
| LogMelSpectrogram | (125, 229, 1) | - | - | - | - |
| conv2d | (125, 115, 64) | 7 | 7 | 2 | 64 |
| max pool | (125, 58, 64) | 1 | 3 | 2 | - |
| residual block | (125, 58, 128) | 3 | 3 | 1 | 128 |
| residual block | (125, 57, 128) | 3 | 3 | 1 | 128 |
| residual block | (125, 29, 256) | 3 | 3 | 2 | 256 |
| residual block | (125, 29, 256) | 3 | 3 | 1 | 256 |
| residual block | (125, 29, 256) | 3 | 3 | 1 | 256 |
| residual block | (125, 15, 512) | 3 | 3 | 2 | 512 |
| residual block | (125, 15, 512) | 3 | 3 | 1 | 512 |
| residual block | (125, 15, 512) | 3 | 3 | 1 | 512 |
| residual block | (125, 15, 512) | 3 | 3 | 1 | 512 |
| residual block | (125, 8, 1024) | 3 | 3 | 2 | 1024 |
| residual block | (125, 8, 1024) | 3 | 3 | 1 | 1024 |
| residual block | (125, 8, 1024) | 3 | 3 | 1 | 1024 |
| dense | (125, 1, 128) | - | - | 128 | 1 |
| upsample time | (1000, 1, 128) | - | - | - | - |
| softplus and normalize | (1000, 1, 128) | - | - | - | - |

Table 2: Model architecture for the f(t) encoder using a Resnet on log mel spectrograms. Spectrograms have a frame size of 2048 and a hop size of 512, and are upsampled at the end to have the same time resoultion as other latents (4ms per a frame). All convolutions use "same" padding and a temporal stride of 1. Each residual block uses a bottleneck structure (He et al., 2016). The final output is a normalized probablity distribution over 128 frequency values (logarithmically scaled between 8.2Hz and 13.3kHz (`https://www.inspiredacoustics.com/en/MIDI_note_numbers_and_center_frequencies`)). The finally frequency value is the weighted sum of each frequency by its probability.

## B.2 DECODER

The decoder's input is the latent tuple $(f(t), l(t), z(t))$ (250 timesteps). Its outputs are the parameters required by the synthesizers. For example, in the case of the harmonic synthesizer and filtered noise synthesizer setup, the decoder outputs $a(t)$ (amplitudes of the harmonics) for the harmonic synthesizer (note that $f(t)$ is fed directly from the latent), and $H$ (transfer function of the FIR filter) for the filtered noise synthesizer.

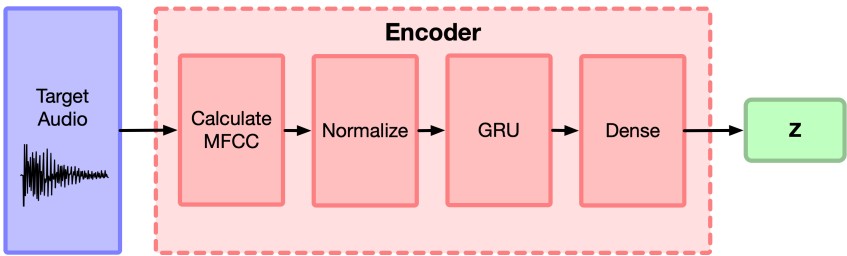

Figure 8: Diagram of the $z$-encoder.

As shown in Figure 9, we use a "shared-bottom" architecture, which computes a shared embedding from the latent tuple, and then have one head for each of the $(\boldsymbol{a}(t), \boldsymbol{H})$ outputs.

In particular, we apply separate MLPs to each of the $(f(t), l(t), \boldsymbol{z}(t))$ input. The outputs of the MLPs are concatenated and passed to a 512-unit GRU. We concatenate the GRU outputs with the outputs of the $f(t)$ and $l(t)$ MLPs (in the channel dimenssion) and pass it through a final MLP and Linear layer to get the decoder outputs.

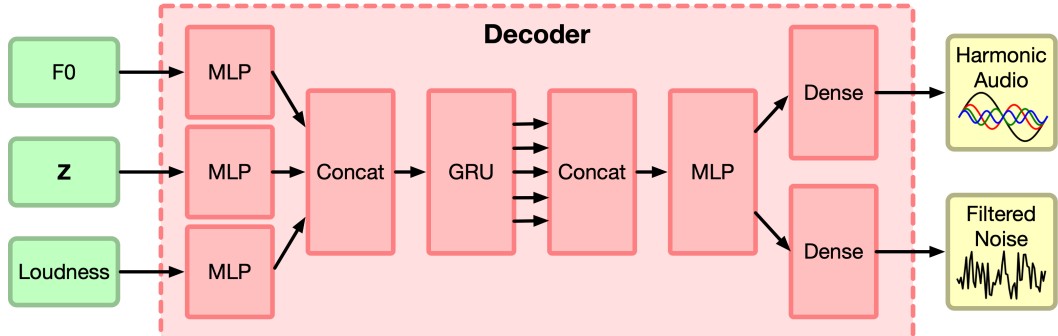

Figure 9: Diagram of the decoder for the harmonic synthesizer and the filtered noise synthesizer.

The MLP architecture, shown in Figure 10, is a standard MLP with a layer normalization (`tf.contrib.layers.layer_norm` ) before the RELU nonlinearity. In Figure 9, all the MLPs have 3 layers and each layer has 512 units.

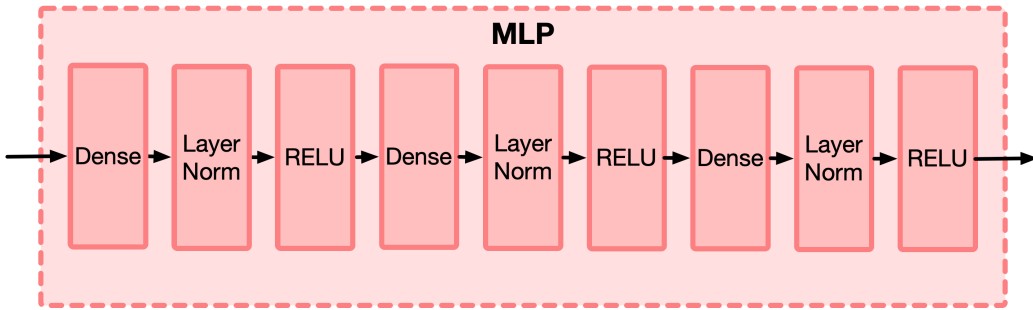

Figure 10: MLP in the decoder.

## B.3 TRAINING

Because all DDSP components are differentiable, the model is differentiable end-to-end. Therefore, we can apply any SGD optimizer to train the model. We used ADAM optimizer with learning rate 0.001 and exponential learning rate decay 0.98 every 10,000 steps.

### B.4 PERCEPTUAL LOSS

To help guide the DDSP autoencoder that must predict $f(t)$ on the NSynth dataset, we also added an additional perceptual loss using pretrained models, such as the CREPE (Kim et al., 2018) pitch estimator and the encoder of the WaveNet autoencoder (Engel et al., 2017). Compared to the L1 loss on the spectrogram, the activations of different layers in these models correlate better with the perceptual quality of the audio. After a large-scale hyperparameter search, we obtained our best results by using the L1 distance between the activations of the small CREPE model's fifth max pool layer with a weighting of $5 \times 10^{-5}$ relative to the spectral loss.

### B.5 SYNTHESIZERS

**Harmonic synthesizer / Additive Synthesis**: We use 101 harmonics in the harmonic synthesizer (i.e., $a(t)$'s dimension is 101). Amplitude and harmonic distribution parameters are upsampled with overlapping Hamming window envelopes whose frame size is 128 and hop size is 64. Initial phases are all fixed to zero, as neither the spectrogram loss functions or human perception are sensitive to absolute offsets in harmonic phase. We also do not include synthesis elements to model DC components to signals as they are inaudible and not reflected in the spectrogram losses.

We force the amplitudes, harmonic distributions, and filtered noise magnitudes to be non-negative by applying a sigmoid nonlinearity to network outputs. We find a slight improvement in traning stability by modifying the sigmoid to have a scaled output, larger slope by exponentiating, and threshold at a minimum value:

$$y = 2.0 \cdot \text{sigmoid}(x)^{\log 10} + 10^{-7} \tag{5}$$

**Filtered noise synthesizer**: We use 65 network output channels as magnitude inputs to the FIR filter of the filtered noise synthesizer.

### B.6 PARAMETER COUNTS

| Model | Parameters |
|---|---|
| WaveNet Autoencoder (Engel et al., 2017) | 75M |
| WaveRNN (Hantrakul et al., 2019) | 23M |
| GANSynth (Engel et al., 2019) | 15M |
| DDSP Autoencoder (Unsupervised) | 12M |
| DDSP Autoencoder (Supervised, NSynth) | 7M |
| DDSP Autoencoder (Supervised, Solo Violin) | 6M |
| DDSP Autoencoder Tiny (Supervised, Solo Violin) | 0.24M |

Table 3: Parameter counts for different models. All models trained on NSynth dataset except for those marked (Solo Violin). Autoregressive models have the most parameters with GANs requiring less. The DDSP models examined in this paper (which have not been optimized at all for size) require 2 to 3 times less parameters than GANSynth. The unsupervised model has more parameters because of the CREPE (small) $f(t)$ encoder, and the autoencoder has additional parameters for the $z(t)$ encoder. Initial experiments with *extremely* small models (single GRU, 256 units), have slightly less realistic outputs, but still relatively high quality (as can be heard in the supplemental audio).

## C EVALUATION DETAILS

### C.1 METRICS

**Loudness $L_1$ distance**: The loudness vector is extracted from the synthesized audio and $L_1$ distance computed against the input's conditioning loudness vector (ground truth). A better model will produce lower $L_1$ distances, indicating input and generated loudness vectors closely match. Note this distance is *not* back-propagated through the network as a training objective.

**F0 $L_1$ distance**: The F0 $L_1$ distance is reported in MIDI space for easier interpretation; an average F0 $L_1$ of 1.0 corresponds to a semitone difference. We use the same confidence threshold of 0.85 in (Hantrakul et al., 2019) to select portions where there was detectable pitch content, and compute the metric only in these areas.

**F0 Outliers:** Pitch tracking using CREPE, like any pitch tracker, is not completely reliable. Instabilities in pitch tracking, such as sudden octave jumps at low volumes, can result errors not due to model performance and need to be accounted for. F0 outliers accounts for pitch tracking imperfections in CREPE (Kim et al., 2018) vs. genuinely bad samples generated by the trained model. CREPE outputs both an F0 value as well as a F0 confidence. Samples with confidences below a threshold of 0.85 in (Hantrakul et al., 2019) are labeled as outliers and usually indicate the sample was mostly noise with no pitch or harmonic component. As the model outputs better quality audio, the number of outliers decrease, thus lower scores indicate better performance.

## C.2 INTERPOLATION METRICS

Loudness $L_1$ and F0 $L_1$ are shown for different interpolation tasks in table C.2. In reconstruction, the model is supplied with the standard $(f(t)_A, l(t)_A, \boldsymbol{z}(t)_A)$. Loudness $(L_1)$ and F0 $(L_1)$ are computed against the ground truth inputs. In loudness interpolation, the model is supplied with $(f(t)_A, l(t)_B, \boldsymbol{z}(t)_A)$, and Loudness $(L_1)$ is calculated using $l(t)_B$ as ground truth instead of $(l(t)_A)$. For F0 interpolation, the model is supplied with $(f(t)_B, l(t)_A, \boldsymbol{z}(t)_A)$ and F0 $(L_1)$ is calculated using $f(t)_B$ as ground truth instead of $f(t)_A$. For Z interpolation, the model is supplied with $(f(t)_A, l(t)_A, \boldsymbol{z}(t)_B)$. The low and constant Loudness $(L_1)$ and F0 $(L_1)$ metrics across these interpolations indicate the model is able independently vary these variables without affecting other components.

| Task | Loudness $(L_1)$ | F0 $(L_1)$ |
|---|---|---|
| Reconstruction | 0.042 | 0.060 |
| Loudness $l(t)$ interp. | 0.061 | 0.060 |
| F0 $f(t)$ interp. | 0.048 | 0.070 |
| Z $z(t)$ interp. | 0.063 | 0.065 |

Table 4: Loudness and F0 metrics for different interpolation tasks.

# D    OTHER NOTES

## D.1    SINUSOIDAL MODELS

While unconstrained sinusoidal oscillator banks are strictly more expressive than harmonic oscillators, we restricted ourselves to harmonic synthesizers for the time being to focus the problem domain. However, this is not a fundamental limitation of the technique and perturbations such as inharmonicity can also be incorporated to handle phenomena such as stiff strings (Smith, 2010).

## D.2    REGULARIZATION

It is worth mentioning that the modular structure of the synthesizers also makes it possible to define additional losses in terms of different synthesizer outputs and parameters. For example, we may impose an SNR loss to penalize outputs with too much noise if we know the training data consists of mostly clean data. We have not experimented too much with such engineered losses, but we believe they can make training more efficient, even though such engineering methods deviates from the end-to-end training paradigm,

