# OpenReview forum: "DDSP: Differentiable Digital Signal Processing"
_ICLR.cc/2020/Conference — Accept (Spotlight)_

### Official Review · AnonReviewer1 · 2019-10-28
**Official Blind Review #1**

**Rating:** 8

**Review:**

This paper develops a framework for for audio generation using oscillators with differentiable neural network type learning. They showcase the usefulness and effectiveness of the approach with several examples such as timbre transfer, dereverberation, changing the room impulse response, pitch extrapolation and so on. I can imagine the proposed learnable oscillator based autoencoders in a variety of applications.

I think that this suggested software library can be useful for a wide range of audio researchers, and I commend the authors for this contribution. It is very nice to see an example of research where we make use of our physical understanding of the sound medium rather than blindly throwing a neural network at the problem.

I have one important question though: how susceptible do you think the system is robust with respect to f0 and loudness encoders? Have you experimented with situations where the f0 and the loudness encoders might fail (such as more non-periodic and noisy signals)?


**Experience Assessment:**

I have read many papers in this area.

**Review Assessment: Checking Correctness Of Derivations And Theory:**

I assessed the sensibility of the derivations and theory.

**Review Assessment: Checking Correctness Of Experiments:**

I assessed the sensibility of the experiments.

**Review Assessment: Thoroughness In Paper Reading:**

I read the paper at least twice and used my best judgement in assessing the paper.

---

> ### Author Response · Authors · 2019-11-13
> **Responses to AnonReviewer1**
>
> Thank you for your review and helpful comments. We have replied to your main question below.
>
> > “how susceptible do you think the system is robust with respect to f0 and loudness encoders? Have you experimented with situations where the f0 and the loudness encoders might fail (such as more non-periodic and noisy signals)?”
>
> For the specific harmonic+noise model we consider in this paper, accurate f0 estimation is very important. It is a strong constraint of the model by construction. We can see this when we train a DDSP autoencoder with a learnable f0 encoder. The model first reduces loss by covering the spectrogram with filtered noise, and only later replaces that noise with harmonic content when it can more accurately follow the f0 contours. The loudness conditioning signal is extracted directly from the audio, so is not a source of such variability.
>
> Due to its construction, the harmonic constraints are not appropriate for modeling non-periodic signals, and we haven’t tested on that type of data. However, there are many similar variants (such as an unconstrained sinusoidal+residual model) that have been shown to work well on more general-purpose sounds, and this is definitely an area we’d like to explore in future research.
>
> Beyond sinusoidal+noise modeling, we found in our experiments that waveforms can be added linearly. This means using DDSP components does not preclude using raw waveform generation. We demonstrate this in the paper by adding the output waveforms of the additive and noise synthesizers, but future work can extend this waveforms generated directly by neural networks, which may be an efficient manner of representing transients.

---

### Official Review · AnonReviewer3 · 2019-10-28
**Official Blind Review #3**

**Rating:** 6

**Review:**

This paper presents a model for audio generation/synthesis  where the model is trained to output time-varying parameters of a vocoder/synthesiser rather than directly outputting audio samples/spectrogram frames. The model is trained by mining an L1 loss between the synthesised audio and the real training audio. Overall, I found the paper to be well written, I found the online supplementary material helpful in getting an intuition for the quality of audio samples generated and to understand the various parts of the proposed architecture. I also found the figures to be extremely  informative, especially figure 2. Although, I think the title of the paper could be updated to something more specific like Differentiable Vocoders or something similar, since the description and experiments very specifically deal with audio synthesis with vocoders, even though the components might be general DSP components. I think the paper presents a reasonable alternative to current autoregressive audio generation methods and should be accepted for publication.

Minor Comments

1. The reference provided for RNNs, Sutskever et. al. 2014, should be supplemented with older references from the 80s when RNNs were first trained with backdrop through time.
2. “The bias of the natural world is to vibrate.” I am not sure what exactly is meant here. Is it really a “bias” if every object in the universe vibrates? I think the term bias has been overloaded in the paper and is confused with structural/inductive priors. Bias as used in the paper, seems to have several means. It would help the description if the authors cleared up this confusing use of the term.
3. In Section 1.3, the authors claim that one of the strengths of the proposed method is that the models are “relatively small”, however all discussion of the sizes is relegated to the appendix. It would be helpful to the reader if some model complexity comparisons are presented in the results section and a high level summary is presented at the end of Section 1.
4. In Section 3, some additional high-level description for the Harmonic plus Noise model (Serra & Smith, 1990) should be provided to motivate the discussion and experiments in the rest of the paper.
5. Section 4.2 should provide some description of the parameters counts for each of the components considered and how this compares with existing auto-regressive generation algorithms.


**Experience Assessment:**

I have read many papers in this area.

**Review Assessment: Checking Correctness Of Derivations And Theory:**

I assessed the sensibility of the derivations and theory.

**Review Assessment: Checking Correctness Of Experiments:**

I assessed the sensibility of the experiments.

**Review Assessment: Thoroughness In Paper Reading:**

I read the paper thoroughly.

---

> ### Author Response · Authors · 2019-11-13
> **Responses to AnonReviewer3 (part 1/2)**
>
> Thank you for your time and expertise in your review. We’ve done our best to address your key points with paper revisions and the comments below.
>
> > “I think the title of the paper could be updated to something more specific like Differentiable Vocoders or something similar, since the description and experiments very specifically deal with audio synthesis with vocoders, even though the components might be general DSP components.”
>
> This is a very valid point and something we’ve done a lot of back and forth about. A more limited title would be more descriptive of the specific experiments performed in the paper. However, a main partner of the paper is the release of the corresponding Tensorflow library, that we hope will see use in a much broader range of applications. As the library will likely not receive its own paper in another venue, we would like this paper to serve as the initial and primary reference, which the current title accomplishes. It’s a tough call, but I think we’d like to stick with the title as it is.
>
> > “1. The reference provided for RNNs, Sutskever et. al. 2014, should be supplemented with older references from the 80s when RNNs were first trained with backdrop through time.”
>
> Agreed. We’ve added citations to a popular paper by Werber and book by Williams and Zipser from 1990.
>
> > “2. “The bias of the natural world is to vibrate.” I am not sure what exactly is meant here. Is it really a “bias” if every object in the universe vibrates? I think the term bias has been overloaded in the paper and is confused with structural/inductive priors. Bias as used in the paper, seems to have several means. It would help the description if the authors cleared up this confusing use of the term.”
>
> Thank you for pointing out that we’ve overloaded the term “bias”, which we indeed use interchangeably with structural/inductive priors. We’ve amended the paper in several places to be more explicit. For instance we’ve rewritten the paragraph in the introduction to try and be more straightforward:
>
> “””
> Objects have a natural tendency to periodically vibrate. Small shape displacements are usually restored with elastic forces that conserve energy (similar to a canonical mass on a spring), leading to harmonic oscillation between kinetic and potential energy (Smith, 2010).... ...However, neural synthesis models often do not exploit this periodic structure for generation and perception.
> “””
>
> > “4. In Section 3, some additional high-level description for the Harmonic plus Noise model (Serra & Smith, 1990) should be provided to motivate the discussion and experiments in the rest of the paper.”
>
> We agree this would strengthen the paper, as the model expressivity seems to have been a point of confusion for some readers. We have added a Section 3.1 with citations to better motivate the use and expressivity of the model:
>
> “””3.1
> SPECTRAL MODELING SYNTHESIS
> Here, as an example DDSP model, we implement a differentiable version of Spectral Modeling Synthesis (SMS) Serra & Smith (1990). This model generates sound by combining an additive synthesizer (adding together many sinusoids) with a subtractive synthesizer (filtering white noise). We choose SMS because, despite being parametric, it is a highly expressive model of sound, and has found widespread adoption in tasks as diverse as spectral morphing, time stretching, pitch shifting, source separation, transcription, and even as a general purpose audio codec in MPEG-4 (Tellman et al., 1995; Sanjaume, 2002; Klapuri et al., 2000; Purnhagen & Meine, 2000).
>
> As we only consider monophonic sources in these experiments, we use the Harmonic plus Noise model, that further constrains sinusoids to be integer multiples of a fundamental frequency (Beauchamp, 2007). One of the reasons that SMS is more expressive than many other parametric models because it has so many more parameters. For example, in the 4 seconds of 16kHz audio in the datasets considered here, the synthesizer coefficients actually have ∼2.5 times more dimensions than the audio waveform itself ((1 amplitude + 100 harmonics + 65 noise band magnitudes) * 1000 timesteps = 165,000 dimensions, vs. 64,000 audio samples). This makes them amenable to control by a neural network, as it would be difficult to realistically specify all these parameters by hand.
> “””

---

> > ### Author Response · Authors · 2019-11-13
> > **Responses to AnonReviewer3 (continued 2/2)**
> >
> > > “ ...the authors claim that one of the strengths of the proposed method is that the models are “relatively small”... ...Section 4.2 should provide some description of the parameters counts for each of the components considered and how this compares with existing auto-regressive generation algorithms.”
> >
> > Thank you for highlighting that we should compare model sizes more explicitly, as we agree that it would help support the paper. Accordingly, we have added a new table in the supplemental with parameter counts from all comparable models. We have also conducted some promising initial experiments in reducing model size that we have added to the supplement. We have correspondingly added the following paragraph to Section 4.1 (which has been swapped with section 4.2 for better flow):
> >
> > “””
> > Model Size:
> > Table B.6, compares parameter counts for the DDSP models and comparable models including GANSynth (Engel et al., 2019), WaveRNN (Hantrakul et al., 2019), and a WaveNet Autoencoder (Engel et al., 2017). The DDSP models have the fewest parameters (up to 10 times less), despite no effort to minimize the model size in these experiments. Initial experiments with very small models (240k parameters, 300x smaller than a WaveNet Autoencoder) have less realistic outputs than the full models, but still have fairly high quality and are promising for low-latency applications, even on CPU or embedded devices. Audio samples are available in the online supplement.
> > “””
> >
> > Reference to the size comparisons has also been added in the introduction under the part for autoregressive models being very large.

---

### Official Review · AnonReviewer2 · 2019-11-03
**Official Blind Review #2**

**Rating:** 8

**Review:**

This very nice paper tackles the integration of of domain knowledge in signal processing within neural nets; the approach is illustrated in the domain of music.

The proof of concept (audio supplementary material) is very convincing.

The argument is that a "natural" latent space for audio is the spectro-temporal domain. Approaches working purely in the waveform, or in the frequency domains must handle the phase issues. Approaches can learn to handle these issues, at the expense of more data.
A key difficulty is that the L_2 loss does not match the perception. The authors present a perceptual loss that addresses the point (Eq. 4 - clarify the difference w.r.t. Wang et al.), . A natural question thus is whether applying this loss on e.g. Wavenet, Sample RNN or Wave RNN would solve the problem.

In short, the contribution is in designing a latent space that accounts for independent components of the music signal (pitch; loudness; reverberation and/or noise), using existing components (oscillators, envelopes and filters), and making them amenable to end-to-end optimization (noting that loudness can be extracted deterministically).

I understand that the auto-encoder is enriched with a FIR filter at its core: the input is mapped into the time-frequency domain; convolved with the output of the neural net H_l, and the result is recovered from the time-frequency domain.
Explain "frames x_l to match the impulse responses h_l".

Care (domain knowledge and trials and errors, I suppose) is exercized in the conversion and recovery (shape and size of the window) to remove undesired effects.

Overall, the approach works in two modes: one where the fundamental frequency is extracted, one where it is learned. I miss this comparison in the audio material, could you tell where to look / hear ?


Questions:
* NN operate at a slower frame rate (sect. 3.2): how much slower ? How sensitive to this parameter ?

Detail
* were, end p. 5. A word missing ?
* are useful --> is useful ?
* could produced, p.6

**Experience Assessment:**

I have read many papers in this area.

**Review Assessment: Checking Correctness Of Derivations And Theory:**

N/A

**Review Assessment: Checking Correctness Of Experiments:**

I assessed the sensibility of the experiments.

**Review Assessment: Thoroughness In Paper Reading:**

I read the paper thoroughly.

---

> ### Author Response · Authors · 2019-11-13
> **Responses to AnonReviewer2**
>
> Thank you for the review and the encouraging comments. We’ve done our best to address your questions with paper revisions and the comments below.
>
> > “The authors present a perceptual loss that addresses the point (Eq. 4 - clarify the difference w.r.t. Wang et al.), . A natural question thus is whether applying this loss on e.g. Wavenet, Sample RNN or Wave RNN would solve the problem.”
>
> The loss in Equation 4 is a multi-scale spectrogram loss in both magnitude and log-magnitude. Wang et al. (2019) use a similar loss. Their loss differs in that they use different window/hop sizes than this paper, 3 scales instead of 6, and a phase loss instead of a linear magnitude loss. As described in Section B.4, we also use a “perceptual” loss for the unsupervised DDSP autoencoder (that must jointly learn to infer f(t)), using the activations of a pretrained CREPE model.
>
> We agree that it would be great to be able to use the multi-scale spectrogram losses and pretrained model losses for autoregressive waveform models such as WaveNet, SampleRNN, or WaveRNN. Unfortunately, one of the drawbacks of these models (as described in Section 1) is that they are trained with teacher forcing and do not generate samples during training (which would be computationally infeasible). This prevents using any losses that actually compare target audio and generated audio, and motivates the approach taken in this paper (and others e.g. GANSynth, NSF, etc.).
>
>
> > “I understand that the auto-encoder is enriched with a FIR filter at its core: the input is mapped into the time-frequency domain; convolved with the output of the neural net H_l, and the result is recovered from the time-frequency domain. Explain "frames x_l to match the impulse responses h_l".”
>
> For memory/compute efficiency, the outputs of the neural network are not at audio rate. In the paper experiments the FIR coefficients are provided every 4ms i.e. for audio at 16kHz, the FIR coefficients are provided every 64 samples.
>
> To make this concrete, the size of the coefficients tensor is then (n_batch, n_frames=1000, n_coefficients=65), while the size of audio is (n_batch, n_samples=64000). The audio is then divided into the same number of frames, (n_batch, n_frames=1000, n_samples_per_frame=64), using non-overlapping box windows. We then take the DFT of each frame, multiply them with IDFT(Window(DFT(coefficients)) and take the IDFT to get the filtered audio frames (n_batch, n_frames=1000, n_samples_per_frame=64). Finally we flatten the frames (box window with no overlap) to get the filtered audio (n_batch, n_samples=64000).
>
> > “Overall, the approach works in two modes: one where the fundamental frequency is extracted, one where it is learned. I miss this comparison in the audio material, could you tell where to look / hear ?”
>
> Good point. We have added comparisons of the reconstructions to the online supplement.
>
> > “NN operate at a slower frame rate (sect. 3.2): how much slower ? How sensitive to this parameter ?”
>
> As we mentioned above, the network in these experiments operates at 1 frame every 4ms (64 samples). We’ve also run experiments with 1 frame every 16ms (256 samples), which are actually faster to optimize, but are limited in the temporal response of quick attacks.
>
> > “were, end p. 5. A word missing? are useful --> is useful? could produced, p.6”
>
> Thanks for the help catching the typos! We’ve fixed them in the revised version of the manuscript.

---

### Comment · Area_Chair1 · 2019-11-06
**some concerns about the proposed autoencoder architecture**

A signal processing framework, called DDSP, is proposed. DDSP is based on an autoencoder architecture. Three different encoders are adopted to represent the latent coders, while a deterministic synthesizer is used to fusion the latent coder to do reconstruction. Replacing the encoder/decoder with some deterministic module is a popular approach to pursue interpretability. And the results along with the online audio are very impressive.

However, I have some concerns about the proposed autoencoder architecture:

(1) The authors replaced the decoder with some deterministic synthesizer module, which would constrain the model expressive power due to the low complexity of the predefined module.

(2) An MLP decoder is used along with simple concatenation. The output is then used for the harmonic synthesizer and the filtered noise synthesizer. It would fail to get any meaningful interpretability for the latent code since all latent encoder are highly entangled.

(3) It is highly claimed in the paper that the proposed DDSP shows potential in many interesting tasks, while insufficient experiment results are shown. The corresponding experiment results are needed to support the claims in the paper.

(4) It is claimed in Section 1.2 that "small errors in parameters can lead to large errors in the audio that cannot propagate back to the network". I am very curious about how this issue is addressed or elevate in this paper since a similar deterministic synthesizer module in DDSP.

(5) Too many contents are left in the supplemental. Section 3 only details the formula of the yellow component of Figure 2, while other components are omitted. It fails to give a clear description of the whole DDSP work.

(6) In the supplemental, it is claimed that the f(t) is replaced with a fixed network during the training, which is different from the claim in the main body that f(t) is the output of the encoder. Does it mean the direct neural encoder f(t) is not a good option?

(7) As claimed in the paper, f(t) is a fixed network, while z is not used in the model trained on solo violin. It means all the encoder parts is no longer exist. Does it mean the decoder is the main component of DDSP?

(8) Regarding the multi-scale spectral loss in Eq.(4), a trade off parameter is needed to balance the two L1 loss.

---

> ### Author Response · Authors · 2019-11-13
> **Responses to Area Chair1 (part 1/3)**
>
> We thank the area chair for their helpful comments and for calling attention to some aspects of our paper that could benefit from clarification. We have done our best to address each comment independently below and amend the paper where appropriate.
>
> From a high-level, most of the confusion seems to stem from the perspective in Comment (1) that “DDSP is based on an autoencoder architecture”. As we discuss below, DDSP is the library of differentiable Digital Signal Processing (DSP) components, agnostic to the given network architecture. It is important to consider DDSP components as analogous to network components (such as convolutional layers or recurrent layers) rather than a specific generative model itself.
>
> To further clarify, we have added the following paragraph to Section 4.1 where we introduce the autoencoder architecture:
> “”””
> DDSP components do not put constraints on the choice of generative model (GAN, VAE, Flow, etc.), but we focus here on a deterministic autoencoder to investigate the strength of DDSP components independent of any particular approach to adversarial training, variational inference, or Jacobian design. Just as autoencoders utilizing convolutional layers outperform fully-connected autoencoders on images, we find DDSP components are able to dramatically improve autoencoder performance in the audio domain. Introducing stochastic latents (such as in GAN, VAE, and Flow models) will likely further improve performance, but we leave that to future work as it is orthogonal to the core question of DDSP component performance that we investigate in this paper.
> “””
>
>
> >” (1) The authors replaced the decoder with some deterministic synthesizer module, which would constrain the model expressive power due to the low complexity of the predefined module.”
>
> This is an important point, thank you for highlighting that this could use further clarification. Many parametric synthesis models have constrained expressive power because they seek to reduce the signal to only a few parameters. However, the Sinusoidal plus Noise model considered in this paper is very expressive because it actually has many parameters, and has been used in diverse applications, including even general-purpose audio compression in the MPEG-4 standard[1].
>
> To make this clearer to the reader we have added the following section 3.1 to the text:
>
> “””3.1
> SPECTRAL MODELING SYNTHESIS
> Here, as an example DDSP model, we implement a differentiable version of Spectral Modeling Synthesis (SMS) Serra & Smith (1990). This model generates sound by combining an additive synthesizer (adding together many sinusoids) with a subtractive synthesizer (filtering white noise). We choose SMS because, despite being parametric, it is a highly expressive model of sound, and has found widespread adoption in tasks as diverse as spectral morphing, time stretching, pitch shifting, source separation, transcription, and even as a general purpose audio codec in MPEG-4 (Tellman et al., 1995; Sanjaume, 2002; Klapuri et al., 2000; Purnhagen & Meine, 2000).
>
> As we only consider monophonic sources in these experiments, we use the Harmonic plus Noise model, that further constrains sinusoids to be integer multiples of a fundamental frequency (Beauchamp, 2007). One of the reasons that SMS is more expressive than many other parametric models because it has so many more parameters. For example, in the 4 seconds of 16kHz audio in the datasets considered here, the synthesizer coefficients actually have ∼2.5 times more dimensions than the audio waveform itself ((1 amplitude + 100 harmonics + 65 noise band magnitudes) * 1000 timesteps = 165,000 dimensions, vs. 64,000 audio samples). This makes them amenable to control by a neural network, as it would be difficult to realistically specify all these parameters by hand.
> “””
>
> As seen in Figure 2, the autoencoder in the paper still uses a neural decoder to generate controls for the DSP components. It is true that it cannot generate arbitrary waveforms, unlike generating directly in the time or frequency domain, but also does not suffer the challenges of phase alignment and spectral leakage detailed in Section 1.
>
> Lastly, as noted in the paper: waveforms can be added linearly. This means using DDSP components does not preclude using raw waveform generation. We demonstrate this in the paper by adding the output waveforms of the additive and noise synthesizers, but future work can extend this waveforms generated directly by neural networks, which may be an efficient manner of representing transients.
>
> [1] https://ieeexplore.ieee.org/document/856031, (download: http://citeseerx.ist.psu.edu/viewdoc/download?doi=10.1.1.118.1019&rep=rep1&type=pdf)

---

> > ### Author Response · Authors · 2019-11-13
> > **Responses to Area Chair1 (continued 2/3)**
> >
> > > “(2) An MLP decoder is used along with simple concatenation. The output is then used for the harmonic synthesizer and the filtered noise synthesizer. It would fail to get any meaningful interpretability for the latent code since all latent encoder are highly entangled.”
> >
> > It is true that, like other deterministic autoencoders, there is no explicit regularization applied to the latent z(t) in this work. However, there is an explicit factorization of z(t) from the other latents (f(t) and l(t)). In practice this results in strong factorization of the effects of each component. As shown in Figure 3 and Table 2, we see quantitative and qualitative evidence that varying f(t) has little effect on the generated loudness, varying l(t) has little effect on the generated frequency, and varying z(t) has little effect on either the generated loudness or frequency. That said, like autoencoders used in image generation, z(t) still captures other aspects of variation even without explicit regularization. We demonstrate this in the audio domain in the right cell of Figure 3, where interpolating z(t) causes the spectral centroid of generated audio to vary smoothly between that of two test samples, while the frequency and loudness contours remain constant.
> >
> >
> > > “(3) It is highly claimed in the paper that the proposed DDSP shows potential in many interesting tasks, while insufficient experiment results are shown. The corresponding experiment results are needed to support the claims in the paper.”
> >
> > The claim of broad applications is based upon the diverse use of traditional Digital Signal Processing components in different applications such as signal processing, communications, image processing, video coding, radar, ultrasound etc. [1]. The DDSP library contains differentiable DSP components (time-varying filters, resampling, oscillators, and others) that mirror their widely used non-differentiable counterparts. To demonstrate the core principle of integrating interpretable DSP with expressive neural networks, we needed to focus on a specific application. We chose audio synthesis, as it is an area that end-to-end learning still struggles for the reasons outlined in Section 1 of the paper.
> >
> > [1] https://en.wikipedia.org/wiki/Digital_signal_processing#Applications
> >
> >
> > > “(4) It is claimed in Section 1.2 that "small errors in parameters can lead to large errors in the audio that cannot propagate back to the network". I am very curious about how this issue is addressed or elevate in this paper since a similar deterministic synthesizer module in DDSP.”
> >
> > The determinism of the DDSP components is orthogonal to this claim about differentiability. Through the experiments in the paper, we demonstrate that neural networks that control DDSP components can be trained in an end-to-end fashion to minimize losses on the generated audio in an unsupervised manner. This contrasts with previous work that directly models synthesizer coefficients (inferred using hand-designed vocoder algorithms) with an autoregressive model or GAN [1].
> >
> > The DDSP approach is not additionally constrained by limitations of hand-designed analysis algorithms to infer the appropriate coefficients for resynthesis, the are learned jointly with audio loss functions and deep networks. Further, evaluating loss functions on generated audio is more aligned to perception than loss functions on synthesizer coefficients.
> >
> > [1] Merlijn Blaauw and Jordi Bonada. A neural parametric singing synthesizer modeling timbre and expression from natural songs. Applied Sciences, 7(12):1313, 2017. (https://arxiv.org/abs/1704.03809)
> >
> >
> > > “(5) Too many contents are left in the supplemental. Section 3 only details the formula of the yellow component of Figure 2, while other components are omitted. It fails to give a clear description of the whole DDSP work.”
> >
> > We agree that it would be preferable to be able to describe the network architecture in more depth in the main text of the paper. However, given space requirements, we focus the main text on the DDSP components (yellow in Figure 2) that make up the library and are the novel contributions of the paper. The network architecture is specifically chosen to be generic (fully connected, deterministic autoencoder) to demonstrate that it is the DDSP components, and not other modeling decisions, that enables the quality of the work.

---

> > > ### Author Response · Authors · 2019-11-13
> > > **Responses to Area Chair1 (continued 3/3)**
> > >
> > > > “(6) In the supplemental, it is claimed that the f(t) is replaced with a fixed network during the training, which is different from the claim in the main body that f(t) is the output of the encoder. Does it mean the direct neural encoder f(t) is not a good option?”
> > >
> > > We apologize for the confusion.  As pointed out, there is a misstatement in the supplementary that claims that we wait for future work to jointly learn the f(t) encoder, when if fact it is done in this work. Thank you for pointing this out and we have significantly reworked Section 4.1 and the supplemental to make this clearer. We now consistently refer to the Supervised DDSP Autoencoder as using a pretrained CREPE model (with fixed weights) for f(t) estimation, while the Unsupervised DDSP Autoencoder uses a Resnet on mel-spectrograms (jointly trained with the rest of the model) to estimate f(t). We have added complete details of the Resnet architecture to the supplemental. Both versions employ neural networks to estimate f(t). Non-neural methods can also be used, but are not currently state-of-the-art at the task.
> > >
> > >
> > > > “(7) As claimed in the paper, f(t) is a fixed network, while z is not used in the model trained on solo violin. It means all the encoder parts is no longer exist. Does it mean the decoder is the main component of DDSP?”
> > >
> > > The main components of DDSP are the differentiable signal processing components controlled by decoder outputs, used in all models (the yellow components in Figure 2). As shown in Figure 2, we denote the decoder as the neural network that controls these components. We draw this distinction to highlight that the DDSP components are agnostic to model architectures and loss function (spectral, adversarial, waveform), as long as they provide the appropriate control signals.
> > >
> > > For clarity, we note that prior to this work, such digital signal processing components have not been implemented in a differentiable form, and could not be trained end-to-end in the manner described here.
> > >
> > > As noted, for the solo violin experiments, we did not include a z encoder. Firstly, this allows us to demonstrate that z(t) is not required for high-quality synthesis, getting good reconstructions from only f(t) and l(t). Unlike the NSynth dataset, the solo violin dataset is a single instrument so z(t) is not needed to account for different instruments. Excluding z(t) also allows us to generalize between domains as in the timbre transfer example. Since we do not explicitly promote cross-domain generalization in z(t) (as in a CycleGAN setup), z(t) wouldn’t be useful for timbre transfer. However, f(t) and l(t) are interpretable by design, allowing extracted features to be applied sensibly in a new domain (the synthesized audio follows f(t) for pitch and l(t) for loudness) as shown in Figure 4.
> > >
> > >
> > > > “(8) Regarding the multi-scale spectral loss in Eq.(4), a trade off parameter is needed to balance the two L1 loss.”
> > >
> > > While hyperparameter tuning of the losses could likely improve results further, in all experiments of this paper, the coefficients each L1 loss is 1.0, so the text of Eq. 4 is technically correct. It is a good point that this is a special case, and we have updated the Eq.4 to have a weighting term (\alpha), and included the sentence: “...where $\alpha$ is a weighting term set to 1.0 in our experiments.”

---

### Author Response · Authors · 2019-11-13
**Paper Updates**

We would like to thank all the reviewers and the area chair for their thoughtful and helpful comments. We have done our best to thoroughly address each one and make appropriate changes to the manuscript. In particular, we would like to highlight several changes in the updated draft

* New table provided in the supplemental with parameter counts for all relevant models.
* Initial results from new “tiny” model (240k parameters) added to the online audio supplement.
* Reworked section 4.1 and 4.2 (DDSP Autoencoder) to more explicitly define the difference between supervised and unsupervised variants.
* Corrections to the f(t) encoder description for unsupervised experiments, including a new table provided in the supplemental with complete architecture details.
* NSynth audio reconstruction comparisons between unsupervised and supervised variants added to the online audio supplement.
* Further motivation provided for the Harmonic plus Noise model in Section 3.1

---

### Public Comment · ~Keunwoo_Choi1 · 2019-12-29
**Questions for reproducing the result**

Hi, thanks for the great work. I have some questions that arose while implementing it. I heard from Hanoi that it's going to be open-sourced very soon, but I thought it'd be helpful anyway to do this Q&A public. I hope it's not overwhelming :) Some of the questions are to be extra sure, but most of them are necessary details that I couldn't get from the paper.

---
Encoder
- How many MFCCs in the encoder?
- So is GRU in the encoder many-to-many?
- Is Dense in the encoder time-distributed dense?
- If the Dense in the encoder a 512-unit linear, how the dim of `z` becomes 16 dim?
- So `z` has 250 time steps, meaning (assuming window size == fft size of 1024), because the hop length is 256, the input length of the target audio (1024 + 256 * 249 = 64798 samples, which is then 4.048 second?)
- (Hantrakul et al., 2019) does not have any Appendix pages. What is the exact formula for the l-encoder?

Decoder
- What's the length of f(t), l(t), z(t)? 250 time steps?
- So are the Dense layers in MLP all time-distributed dense layers?
- "The outputs of MLPs are concatenated" so, each output is (batch, 512, time=250), and the time-step is preserved, making their concat (batch, 1536, 250)?
- "concatenate the GRU outputs" → Does it mean (batch, 512, time=250) is concatenated over time axis, making (batch, 512 * 250=128000)?
- What is the number of hidden units in the final Dense layers?

3.2
- The amplitudes and harmonic distribution -- in other words, are they sequentially i) bilinear-upsampled and then ii) smoothed?

3.3 Filter design
- What is the FFT size, hop length, etc in this module?

3.5 Reverb
- What is the hyperparams of this module? E.g., the reverberation length.

Training
- What's the length/size of the input audio files? 4 second?

---
EDIT: some more questions.
- What happened when there's Z module as well during training on the solo violin dataset? Did it hinder Reverb module to learn the reverberation?

---

> ### Author Response · Authors · 2020-01-31
> **Code now available**
>
> Hi Keunwoo,
>
> Thanks for the helpful questions. We've updated the paper with relevant details, and all further specifics can be found at the GitHub repo: https://github.com/magenta/ddsp
>
> Some real quick responses for posterity.
>
> Encoder
> - How many MFCCs in the encoder?
> >>> 30
>
> - So is GRU in the encoder many-to-many?
> >>> Yes
>
> - Is Dense in the encoder time-distributed dense?
> >>> Yes
>
> - If the Dense in the encoder a 512-unit linear, how the dim of `z` becomes 16 dim?
> >>> It uses a linear layer to compress to 16 dims per a timestep.
>
> - So `z` has 250 time steps, meaning (assuming window size == fft size of 1024), because the hop length is 256, the input length of the target audio (1024 + 256 * 249 = 64798 samples, which is then 4.048 second?)
> >>> Yes, but we trim the end to match.
>
> - (Hantrakul et al., 2019) does not have any Appendix pages. What is the exact formula for the l-encoder?
> >>> https://github.com/magenta/ddsp/blob/master/ddsp/spectral_ops.py#L171
>
> Decoder
> - What's the length of f(t), l(t), z(t)? 250 time steps?
> >>> Yes.
>
> - So are the Dense layers in MLP all time-distributed dense layers?
> >>> Yes.
>
> - "The outputs of MLPs are concatenated" so, each output is (batch, 512, time=250), and the time-step is preserved, making their concat (batch, 1536, 250)?
> >>> Yes.
>
> - "concatenate the GRU outputs" → Does it mean (batch, 512, time=250) is concatenated over time axis, making (batch, 512 * 250=128000)?
> >>> It means they for (batch, 1536, 250) as in the previous question. https://github.com/magenta/ddsp/blob/master/ddsp/training/decoders.py#L60
>
>
> - What is the number of hidden units in the final Dense layers?
> >>> Same, 512.
>
> 3.2
> - The amplitudes and harmonic distribution -- in other words, are they sequentially i) bilinear-upsampled and then ii) smoothed?
> >>> Just bilinearly upsampled. https://github.com/magenta/ddsp/blob/master/ddsp/core.py#L407
>
> 3.3 Filter design
> - What is the FFT size, hop length, etc in this module?
> >>> Hop length is given by the frame size of the filter coefficients (64000 / 250). FFT Size is 512 I believe.
>
> 3.5 Reverb
> - What is the hyperparams of this module? E.g., the reverberation length.
> >>> 64000 samples long, but that's overkill, shorter is fine for most samples.
>
> Training
> - What's the length/size of the input audio files? 4 second?
> >>> For training, yes.
>
> ---
> EDIT: some more questions.
> - What happened when there's Z module as well during training on the solo violin dataset? Did it hinder Reverb module to learn the reverberation?
> >>> We didn't use the Z module during training, to encourage the model to take care of timbre / expression so the user wouldn't have to.

---

### Public Comment · ~Andrey_Kramer1 · 2020-05-10
**Tiny violin model**

Hello, thanks for the library, it's really promising. Could you please share exactly what parameters and corpus were used to train the tiny violin model in this example?
https://storage.googleapis.com/ddsp/index.html#tiny

---

> ### Author Response · Authors · 2020-05-15
> **Re: Tiny violin**
>
> Hi! The gin config for the tiny violin model can now be found in the github repo: https://github.com/magenta/ddsp/blob/master/ddsp/training/gin/papers/iclr2020/tiny_instrument.gin
>
> And the exact pieces in the dataset can be found in the latest draft of the paper (pg.7, footnote 4).

---

### Public Comment · ~Yuchao_Song1 · 2020-08-26
**question regarding the noise synthesizer section**

Great work!

One thing to confirm, in section 3.4, the last sentence, is the $N_l$ IDFT of the noise or DFT? Thanks.

---

> ### Author Response · Authors · 2020-08-26
> **good catch!**
>
> I think you mean section 3.5 right? And it's a good catch, it was a typo and we meant DFT, because we are converting the noise into the frequency domain. We've revised the paper, thanks!

---

> > ### Public Comment · ~Yuchao_Song1 · 2020-08-27
> > **section 3.5**
> >
> > Yes, I meant section 3.5, thanks.

---

### Public Comment · ~Luigi_Attorresi1 · 2021-02-09
**Question on how to model the dataset**

Hello, thank you for sharing your work!
Me and my colleagues are trying to use it in order to model timbres of different sounds found in nature (like animals) and we have a few doubts about how to best design the dataset.

Since it’s very hard to find those sounds with a high variability we have to rely on data augmentation performing pitch scaling and time stretching.
Do you think it would be necessary/helpful to have different samples in terms of loudness and other parameters too?

If the timbre of a sound is very stable and its only occurrences have a fixed pitch and very little dynamics (e.g. car horn) would it be acceptable to have a smaller dataset with respect to those more complex and dynamic timbres for network to learn its characteristics?

Lastly we were wondering if the model would be able to learn complex timbres with a very peculiar envelope or transient like a bird sound.

In addition to that we would like to report that the Colab “timbre_transfer” gives an error when trying to upload a file instead of recording it. We fixed it by changing “audios[0]” with “audios[0][-1]”  in line 16 of the cell Record or Upload Audio.

Thank you in advance!

---

> ### Author Response · Authors · 2021-03-06
> **Re: Question...**
>
> Hi, glad you're finding it useful! For small datasets, any augmentation of pitch and loudness you can do should be helpful as long as it doesn't also dramatically change the timbre of the sound.
>
> Since the model is deterministic, complex timbres are difficult if they require a one to many mapping (one set of pitch/loudness resulting in many timbres). You could get around this by using a generative model such as autoregression or a GAN, but that is not currently in the code base.
>
> Thank you for bringing attention to the bug. There is currently an issue open on the GitHub and we'll hopefully resolve it soon.

---

### Decision · Program_Chairs · 2019-12-19

**Decision:**

Accept (Spotlight)

**Comment:**

This paper proposes a novel differentiable digital signal processing in audio synthesis. The application is novel and interesting. All the reivewers agree to accept it. The authors are encouraged to consider the reviewer's suggestions to revise the paper.